# A Quantile Approach for Retrieving the "Core Urban-Suburban-Rural" (USR) Structure Based on Nighttime Light

**Yaohuan Huang [1,2], Chengbin Wu [1,2,*], Mingxing Chen [1,2], Jie Yang [1,2] and Hongyan Ren [1]**

[1] State Key Laboratory of Resources and Environmental Information System, Institute of Geographic Sciences and Natural Resources Research, Chinese Academy of Sciences, Beijing 100101, China; huangyh@igsnrr.ac.cn (Y.H.); chenmx@igsnrr.ac.cn (M.C.); yangj.20s@igsnrr.ac.cn (J.Y.); renhy@igsnrr.ac.cn (H.R.)

[2] College of Resource and Environment, University of Chinese Academy of Sciences, Beijing 100049, China

\* Correspondence: wucb.19s@igsnrr.ac.cn

**Abstract:** Accurate and timely information on the "core urban-suburban-rural" (USR) spatial structure in a metropolitan region is significant for both the scientific and policy-making communities. However, USR is usually considered as a single land use type, such as an impervious area, rather than three combined subcategories in remote-sensing image retrieval, especially for suburban areas, which obscures the details of the urbanization process. In this paper, we propose a quantile approach to retrieve the structure of USR based on stable nighttime light (NTL) data from the Defense Meteorological Satellite Program/Operational Linescan System (DMSP/OLS) and apply it in the Beijing-Tianjin-Hebei (JJJ) of China from 1995 to 2013. The key parameters of the NTL threshold, which is the maximum change point of the NTL intensity at the USR boundary, used to retrieve the three subcategories of USR are automatically defined based on the quantile approach with three iterations. Then, the overall accuracy and consistency of the retrieval results are evaluated using the corresponding visual interpretation map from Landsat images with a 30 m resolution. Moreover, the influence of parameter uncertainty is compared by introducing the human settlement index (HSI). According to the time-series analysis of USR retrieval in this study, the JJJ experienced rapid urbanization from 1995 to 2013, with the core urban area expanding by 7098 km$^2$ (average increase of 2.7 times), the suburban area expanding by 12,690 km$^2$ (average increase of 2.8 times), and the rural area increasing by 4986 km$^2$ (average increase of 0.38 times). The USR results retrieved based on the approach agree well with the validation of the visual interpretation map, with an overall accuracy (OA) of 0.904 and a kappa coefficient (KC) of 0.650 at the city level. The USR result with the HSI as the input shows that NTL is more suitable for USR structure retrieval as the NTL shows less uncertainty compared with other parameters such as the vegetation index (VI). This study proposes an improved quantile approach for USR mapping from NTL images on a regional scale, which will provide a useful method for urbanization dynamics analysis.

**Keywords:** nighttime lights (NTL); quantile approach; subcategories; core urban-suburban-rural (USR)

## 1. Introduction

Urbanization is widely recognized as an important factor that is one of the grand challenges of humanity, affecting the functions of terrestrial ecosystems, climate change [1], population, medical treatment, policy [2–4], etc. In China, most subcategories of the extent of the urban area display significant "core urban-suburban-rural" (USR) triad structures [4,5]. Recently, rapid Chinese

urbanization caused USR to vary mainly with the processes of expanding the core urban area into the other two subcategories and suburban areas to rural areas. To conduct accurate simulations in various natural and social research, the structure (distribution and proportion) of the USR subcategories of the extent of the urban area has become fundamental data. Furthermore, uncontrolled urban sprawl that many metropolitan areas experience has led to growing USR interface conflicts. The clashes among the interfaces of USR (e.g., the core urban–rural interface, the urban–suburban interface, and the suburban–rural interface) lead to a wide range of land use conflicts [1]. USR land use conflicts have the potential to cause landscape fragmentation, environmental degradation, and even sharp social contradictions, which seem to have again become an object of study for experts [6].

USR has always been considered to be one category of land use such as an impervious surface or residential area, and satellite images have usually been applied as the primary tool for large-scale time-dependent mapping in recent years [7]. Numerous global and regional datasets with various spatial resolutions, such as the EOS/MODIS, USGS, etc. [8–14], can be used for USR extent mapping. Accordingly, a variety of approaches including object-based image analysis (OBIA), visual interpretation classification, decision tree, random forest, expert system, and artificial intelligence neural networks were also applied [7,15–20]. However, the aforementioned datasets and approaches were mainly proposed based on the spectral and textural features of daytime optical satellite images such as medium spatial resolution data (Landsat, MODIS, etc.) and fine resolution satellite data (IKONOS, Quickbird, Gaofen, etc.). In addition, since urban dynamics is a rapid and global process, these multispectral remote-sensing image-based datasets were limited in terms of their temporal coverage and spatial coverage [21,22]. Furthermore, the urban area retrieved from daytime optical satellite images was considered to be an impervious area rather than the combined subcategories of USR, which makes it hard to distinguish the land use subcategories of the extent of the urban area, especially for the suburban area.

Recently, nighttime light (NTL) images have been widely used to retrieve the extent of the urban area [23,24] and have been proven to be effective in the extraction of the extent of the urban area at regional and global scales [25]. NTL images derived from satellite sensors, including the Defense Meteorological Satellite Program/Operational Linescan System (DMSP/OLS), Visible Infrared Imaging Radiometer System (VIIRS), Luojia, etc., have demonstrated the potential for mapping the extent of the urban area due to the benefits of the close relation between NTL and human settlements and the computational efficiency, wide spatial coverage, and relatively long temporal spans of the satellites [21,22]. Therefore, it is significant to map the urban extent at large geographical scales (e.g., continental and global) in a timely and cost-efficient manner. Generally, the approaches for urban extent retrieval from NTL data can be classified into two categories: the threshold-based methods that use only NTL data and the index-based methods that include other satellite data. The former threshold-based approaches map the extent of the urban area by identifying the urban area as pixels whose NTL brightness levels are higher than a predefined threshold (both local and global) [26]. Although a series of studies have demonstrated the reasonable accuracy of threshold-based methods for measuring the extent of the urban area, the methods have problems establishing the optimal global-fixed threshold that varies for the extent of the urban areas with different levels of environmental and socioeconomic development [21,25–27]. Therefore, multiple optimal threshold-based approaches have been widely proposed to segment the extent of the urban area from NTL images [28,29]. Recently, index-based approaches that can remove the weakness [24] of satellite sensors, such as the blooming and saturation effects of DMSP/OLS-NTL, have gained increasing attention due because they provide more detailed information than do NTL data alone for retrieving the extent of the urban area [25,27,30,31]. Among the index-based approaches, the two surface parameters of the vegetation index (VI) and land surface temperature (LST) from different sensors were the most commonly used approaches due to their close relation to urban features [27]. Numerous indices including the human settlement index (HSI) [1], the vegetation adjusted NTL urban index (VANUI) [31], the vegetation temperature light index (VTLI) [32], and the temperature vegetation adjusted NTL urban index (TVANUI) were proposed

by combining NTL and VI, LST, or both. However, a reliable estimate of the optimal thresholds of the indices is still the key parameter for mapping the extent of the urban area, which is the same as in the threshold-based approaches. Besides, as the performance of NTL satellite sensors improve, index-based approaches face the challenge of additional hypotheses between the parameters in an urban area and introducing more data uncertainty among parameters (e.g., the retrieved VI and LST from other satellite images), which may reduce the accuracy of the retrieval of the extent of the urban area.

In light of the aforementioned problems and the fact that most studies focus on retrieving the urban area as a whole due to the "disappearance" of the boundaries among the three USR subcategories, especially suburb [33], we improve a quantile approach for retrieving the extent of USR subcategories from NTL images. Furthermore, an index-based parameter of the HSI is applied as another input in the same study area of the JJJ Metropolitan Area in China to evaluate the accuracy of the improved approach. In this paper, USR corresponds to the core urban, suburban, and rural areas that are defined as the residential land types with obvious distinctions in their NTL intensities.

## 2. Dataset and Study Area

### 2.1. Study Area

The study area of the JJJ, which is both the national capital region of China and largest urbanized region in eastern Asia, lies in North China from 36°N to 42°N and from 111°E to 120°E (as shown in Figure 1). It includes 13 major cities along the coast of China's Bohai Sea. The JJJ has experienced sustained rapid urbanization since the 1980s. In 2014, the Chinese government proposed the Beijing-Tianjin-Hebei Regional Integration (BRI) regional development project to improve the JJJ as a world-class city cluster, which will significantly promote the JJJ's urbanization process [6].

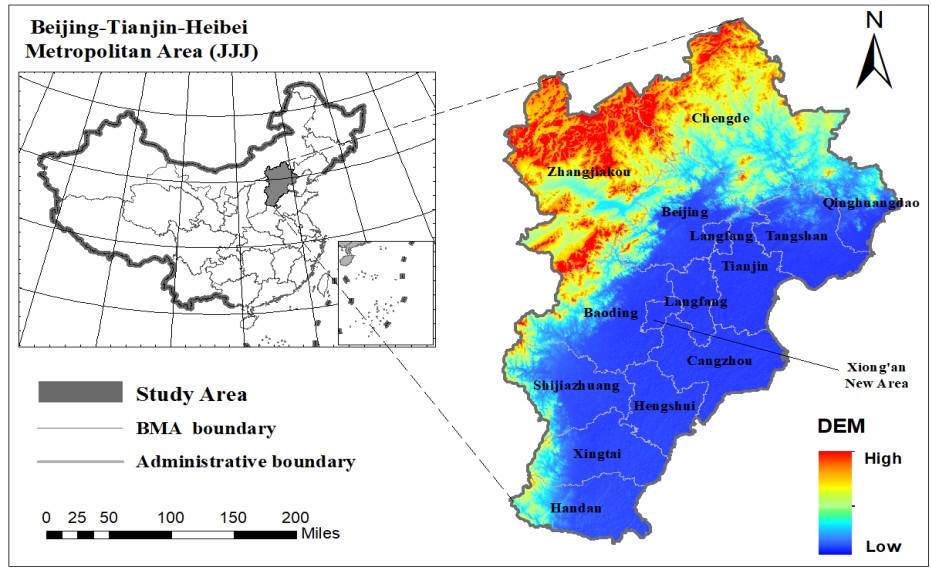

**Figure 1.** Study area (Beijing-Tianjin-Hebei Metropolitan Area, JJJ).

In addition to rapid urbanization over the past several decades, the BRI will inevitably propel the JJJ's urban and population clusters further. However, unplanned urban sprawl requires sufficient land for residential, industrial, and commercial purposes, hospitals, and schools, which generally may consume the precious agricultural and residential land surrounded by cities in the JJJ [34]. Because of the important "Permanent Basic Cropland and Regulation on the Protection of Basic Farmlands" policy in China, suburban areas and rural areas are exposed and particularly vulnerable to core urban area expansion, such as through population suburbanization and industrial transfer to the suburbs. Take the Huilongguan community in Beijing as an example, which used to be a rural area 20 km north

of the core urban area before 2000. The community has been a huge living community in Asia with a population of 0.3 million and an area of 8.5 million m$^2$. Numerous counties surrounding Beijing are also affected by the metropolis' urban expansion, such as the rapid urbanization of eastern Yanjiao Town and southern Gu'an County, which are in Hebei Province. Therefore, the JJJ Metropolitan Area is a typical study area.

## 2.2. Dataset Collection and Preprocessing

The main dataset used in this study for retrieving USR was the DMSP/OLS dataset acquired from (https://ngdc.noaa.gov/eog/download.html). The DMSP/OLS NTL datasets from 1995 to 2013, including F12 (1995–1996), F14 (1997–2000), F15 (2001–2003), F16 (2004–2009), and F18 (2010–2013), were chosen to be consistent with other datasets and the rapid urbanization process of the JJJ. In addition, unstable light sources such as fires and ship lights were excluded from the DMSP/OLS NTL, and water and gas flares were removed based on the MODIS water product MOD44W and gas flare masks [21,35,36] for further analysis. The background (nonlight) values and confounding factors of auroras, fires, boats, and other temporal lights were eliminated or reduced [21]. To calculate the HSI, we used the Terra MODIS NDVI data MOD13A3 in this study, which are calendar-month composites at a 1 km spatial resolution. The quality control flags from MOD13A3 were used to remove abnormal pixels caused by clouds, snow, or other geometric problems. The MOD13A3 datasets of July, August, and September covering the JJJ were chosen, and the best quality image from each month was applied to estimate the HSI. Furthermore, the land use maps of China produced by the Chinese Academy of Sciences were used in our study. Four annual land use datasets from 1995, 2000, 2005, 2010, which were derived from Landsat image with a 30 m resolution based on visual interpretation (the data sets are provided by the Data Center for Resources and Environmental Sciences, Chinese Academy of Sciences (RESDC) http://www.resdc.cn) [23,37], were applied for validation in this study. In this study, based on the 30 m land use data (Appendix A), a set of rural-urban scale verification data was produced manually. To maintain the consistency of spatial resolutions with that of the NTL data, some rural settlements with small areas (almost independent) were excluded. All datasets were processed to a 1 km spatial resolution for consistency and the dataset details are listed in the Table 1.

**Table 1.** Data resources.

| Sensor | Product | Resolution | Acquisition Date | Type |
|---|---|---|---|---|
| DMSP/OLS | Stable light | The annual image product with the grid cell size of 1 km by 1 km | F12: 1995,1996<br>F14: 1997–2001<br>F15: 2002,2003<br>F16: 2004–2009<br>F18: 2010–2013 | Night Light |
| MODIS | MOD44W | Spatial resolution of 1 km for NDVI and EVI. The monthly data was converted into one year data by the maximum method | From 2002 to 2013 | Water Mask |
| MODIS | MOD13A3 | The annual 16 day composite MOD13A2 product with 1-km spatial resolution. Band 1 is NDVI and Band 2 is EVI. | January 2002 to December 2013 (July, August, September) | NDVI and EVI |
| Landsat | 30 meter land use | The per 5 years' data with the grid cell size of 30 m by 30 m. | 1995/2000/2005/2010 | LUCC |

## 3. Methods

Although the current research literature shows a wide variety of sources for USR, there is still no common or consistent definition of this term. In particular, suburbs have been insufficiently considered in the land use retrieval from satellite images. USR is developed based on the flow and redistribution of people, materials, and information among the three subcategories of the extent of the urban area [38]; it essentially reveals the variations of human activities and land use, including the significant differences in NTL. In this study, USR is defined as a residential land use type that consists of core urban, suburban, and rural areas with obvious NTL digital number (DN) values. The general flowchart to delineate the extent of the three CSR subcategories is shown in Figure 2. More details are presented in the following sections.

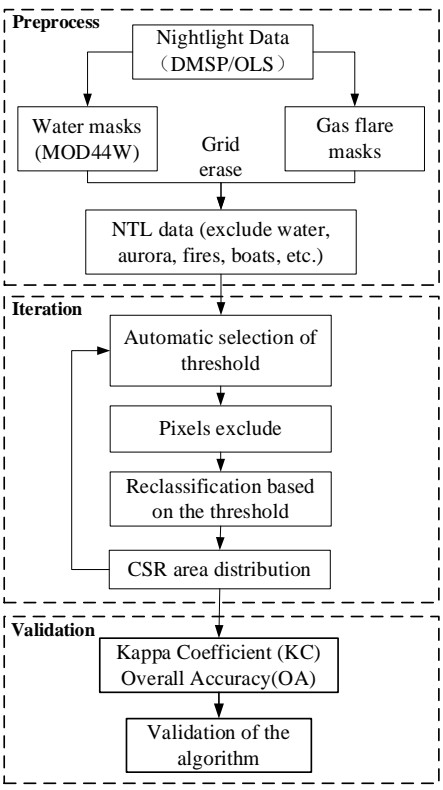

**Figure 2.** Flowchart of the proposed approach.

### 3.1. A Quantile-Based Algorithm for USR Retrieval

In this paper, we improved a multiple iterating quantile approach, which can ignore the complicated impacts of the patterns of USR's spatial distribution and aggregation, to determine the values of the thresholds among three subcategories of USR. The three subcategories of USR were retrieved based on the NTL values from the core urban to rural areas in turn and concentrated in the entire administrative area. Firstly, other areas (e.g., sand, swamp, and undeveloped areas) with values of zero were excluded. Then, all NTL values of the remaining pixels in an administrative area were applied to construct a quantile curve, as shown in Figure 3.

As previously mentioned, optimal thresholds are always important factors when extracting the extent of the urban area from either NTL or related indices. Although the potential extent of USR has various NTL distribution features such as core-urban dominant or the other two subcategories dominant, the key hypothesis of NTL changing rapidly around boundaries among core urban, suburban, and rural areas can be also applied to retrieve the three subcategories of USR (Figure 4).

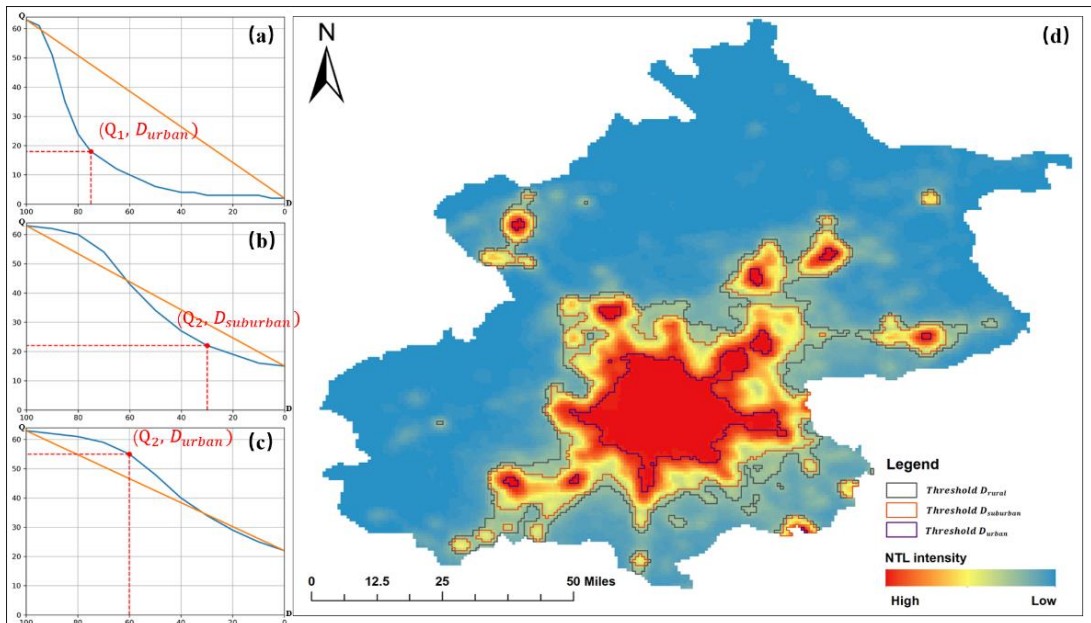

**Figure 3.** Threshold at the urban-suburban-rural (USR) boundary. In (**a**–**c**), the x-coordinate is the quantile (Q), which is arranged inversely from 100th to 0th quantile, and the y-coordinate is the DN value of the NTL DMSP data (D). In (**d**), the threshold corresponds to the USR boundary.

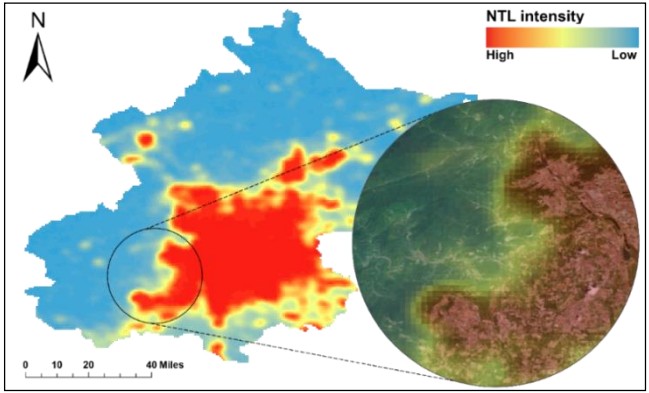

**Figure 4.** NTL intensity transition of the USR boundary.

Based on the assumption that the NTL intensity decreases from urban to rural areas (see Figure 4), the three subcategories of USR were retrieved based on the NTL values from the core urban to rural areas in turn and concentrated in the entire administrative area. In the algorithm, the quantile method selects the threshold from low to high (from rural to urban), and the part of the NTL lower than the threshold is excluded to determine the maximum boundary of the USR subcategories. The NTL data within the maximum boundary are used as the new inputs to continue the next iteration process. When the iterations are finished, the range of the different subcategories of USR can be obtained.

Specifically, quantile curves are constructed by calculating the intensity from percentiles 0–100 in the NTL data and arranging them in reverse order. Taking the line between the starting point and the end point of the curve as the reference line, the point farthest from the reference line is defined as the turning point. The key to retrieve the USR area is to find the NTL intensity corresponding to the turning point via the quantile curve. The calculation process is as follows (Figure 5).

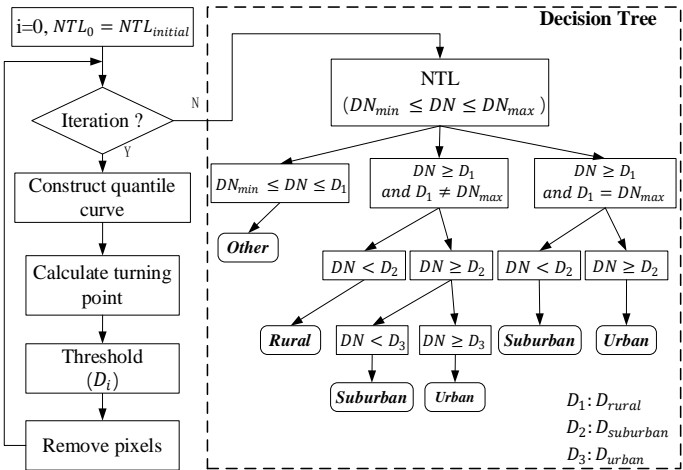

**Figure 5.** USR NTL threshold determination.

In the first iteration, the quantile curve of the NTL data of the whole target area is calculated. Through the distance between the curve and the reference line, the threshold d at the maximum light intensity change in the target area can be obtained (Figure 3a). $D_{rural}$ reflects the sudden change of the NTL from nothing, that is, the border between rural areas and other areas (basically no lighting areas). The part of the NTL data whose NTL intensity is less than $D_{rural}$ is removed to determine the input data of the second iteration.

The second iteration is the same as the first iteration. In the quantile curve constructed in the second iteration, the NTL intensity $D_{suburban}$ (Figure 3b) corresponding to the turning point is the boundary line between the rural and suburban areas. In addition, the part of the input data in the previous step whose NTL intensity is less than $D_{suburban}$ is deleted as the input of the third iteration.

In the third iteration, the input data do not include other regions and rural areas, which is different from the above two iterations. $D_{urban}$ (Figure 3c) has two cases on the quantile curve. When $D_{urban} < DN_{max}$, the corresponding NTL intensity at the turning point, $D_{urban}$, is the boundary between the suburban and urban areas; when $D_{urban} = DN_{max}$, there is no obvious mutation. According to the definition of the NTL intensity of USR, $DN_{urban} > DN_{suburban} > DN_{rural} > DN_{other}$, and the urbanization is not completely reversible in time; therefore, the rural area retrieved in the second iteration should be classified as suburban. The spatial distribution range of USR is obtained by stacking the results of the three iterations.

*3.2. Validation Algorithm*

Visual interpretation of the land use map from the 30 m [37] resolution Landsat multispectral images are recognized as a relatively exact dataset that can be used to validate the accuracy of the improved approach. Then, the 30 m resolution land use data were upscaled to a 1 km spatial resolution to match the resolution of the NTL and HSI data. Moreover, in the existing land use data, suburban and rural areas are usually combined as rural residential land types. We also followed this principle in the evaluation of the results. The two accuracy indices of the kappa coefficient (KC) [39] and overall accuracy (OA) were used in this paper based on calculation of the confusion matrix (as shown in Table 2).

**Table 2.** Confusion matrix of USR (R and S: Rural and suburban area).

| Category in Verification Data | Category in Prediction Data | | |
|---|---|---|---|
| | **Others** | **R and S** | **Urban** |
| **Others** | $C_{11}$ | $C_{12}$ | $C_{13}$ |
| **R and S** | $C_{21}$ | $C_{22}$ | $C_{23}$ |
| **Urban** | $C_{31}$ | $C_{32}$ | $C_{33}$ |

Where $C_{ij}$ is the number of pixles that category *i* in verification data classified as prediction category *j*.

The *KC* is an index used to test the spatial consistency of two maps and can also be used to measure classification accuracy. The *KC* is expressed by

$$KC = \frac{p_o - p_e}{1 - p_e} \tag{1}$$

where $p_o$ and $p_e$ can be defined as:

$$p_o = \frac{\sum_{i=1}^{3} C_{ii}}{n} \tag{2}$$

$$p_e = \frac{\sum_{i=1}^{3}(C_{i,*} \times C_{*,i})}{n^2} \tag{3}$$

where $C_{ii}$ is calculated from Table 2, $C_{i,*}$ is the total number of pixels in the ith row in Table 2, $C_{*,i}$ is the total number of pixels in the ith column in Table 2, and *n* is the total number of pixels in the prediction map from the proposed method.

The value of *KC* is between 0 and 1, which has been used for assessing maps agreement in common research [40,41]. Generally, the *KC* can be divided into five levels [38]: slight (0.0~0.20), fair (0.21~0.40), moderate (0.41~0.60), substantial (0.61~0.80), and almost perfect (0.81~1).

*OA* represents the proportion of the correct categories in the total categories, which can be expressed by

$$OA = p_o = \frac{\sum_{i=1}^{3} C_{ii}}{n} \tag{4}$$

where $C_{ii}$ and *n* are also calculated from Table 2 as Equation (2). In this paper, the *OA* refers to the probability that the USR subcategory is consistent with the 30 m resolution land use data.

## 4. Results

### 4.1. Spatial Distribution of Retrieved USR

Using the above method, we mapped the extent of the USR of the JJJ from 1995 to 2013 based on NTL data (Figure 6). As shown in Figure 6, since 1995, the trend of the urbanization of the JJJ has been obvious, and the extent of the urban area has increased significantly. By 2013, The urban areas of Beijing, Tianjin, and Hebei retrieved by the quantile method were 3253 km$^2$, 1528 km$^2$, and 6720 km$^2$, respectively, (in the 2014 Beijing Land Use Statistical Yearbook, the urban area was 2977.59 km$^2$, and the other two places lack relevant statistical items [42]) and the extent of the urban area has increased by 7098 km$^2$ (an increase of approximately 2.7 times). In addition, the gap between suburban and rural areas has narrowed in terms of the NTL intensity. By 2004, there was no significant difference in the NTL intensity between the suburban and rural areas of Beijing. Tianjin showed the same trend. Around 2006, the gap between the intensity of the NTL in the rural and suburban areas of Tianjin almost disappeared. In Beijing and Tianjin, the rural area increased to approximately 3417 km$^2$ before 2004, and the rural areas gradually merged with the suburban areas after 2004. Different from Beijing and Tianjin, in the process of urbanization in Hebei Province, besides the increase of the urban area by 3955 km$^2$, the rural area also expanded by approximately 8742 km$^2$ (an increase of approximately

0.94 times, higher than the average growth of Beijing and Tianjin), and the distribution was more concentrated and accompanied by a trend of consolidation.

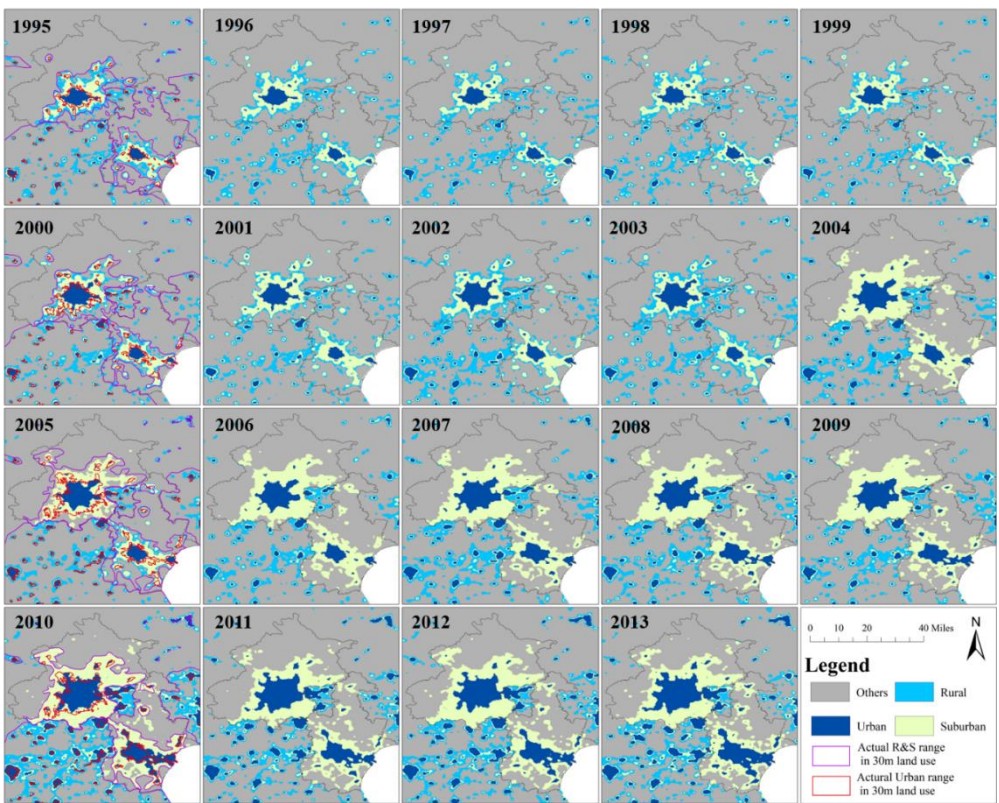

**Figure 6.** USR change in JJJ from 1995 to 2013.

*4.2. Evaluation*

Table 2 shows the classification accuracies for Beijing, Tianjin, and Hebei Province referring to the visual interpretation results. By analyzing the retrieval results in 1995, 2000, 2005, and 2010, it can be seen that the quantile method based on NTL has more stable overall classification accuracy and KC and its effects are slightly different in different regions. The OA and KC in Beijing and Tianjin are significantly higher than those in Hebei. The average OA in Beijing and Tianjin reached 0.904, and the KC averaged 0.650. According to the consistency classification of the KC, the overall classification results of Beijing and Tianjin are highly consistent [39]. In Hebei, the results are generally consistent. When USR is retrieved by NTL, the results are significantly affected by the scale of the target region.

As shown in Table 3, the retrieval accuracies in Beijing and Tianjin are higher than that in Hebei. The differences in the development levels among the cities in Hebei Province are also reflected in the differences in the intensity of NTL. When retrieved by the scope of the entire province, a higher NTL intensity weakens the rural-urban NTL intensity difference in areas with low NTL intensity, and it makes this difference more obvious between different cities in the province. As a result, areas with low NTL are mistakenly classified as rural or suburban areas.

**Table 3.** Accuracies for different cities based on NTL.

| Year | 1995 | | 2000 | | 2005 | | 2010 | | Average | |
|---|---|---|---|---|---|---|---|---|---|---|
| Province | OA | KC | OA | KC | OA | KC | OA | KC | OA | Kappa |
| Beijing | 0.919 | 0.620 | 0.904 | 0.668 | 0.909 | 0.660 | 0.914 | 0.726 | 0.912 | 0.669 |
| Tianjin | 0.926 | 0.638 | 0.859 | 0.591 | 0.900 | 0.623 | 0.895 | 0.672 | 0.895 | 0.631 |
| Hebei | 0.604 | 0.398 | 0.6107 | 0.368 | 0.622 | 0.406 | 0.664 | 0.361 | 0.625 | 0.383 |

In order to compare the retrieval accuracies of different types of land in rural, suburban, and urban areas, we separately calculated the accuracies of core urban land, rural land, suburban land, and others. The results are shown in Table 4.

**Table 4.** Accuracies for different land use types.

| Province | Beijing | | | Tianjin | | | Hebei | | | Average | | |
|---|---|---|---|---|---|---|---|---|---|---|---|---|
| Year | Others | R and S | Urban | Others | R and S | Urban | Others | R and S | Urban | Others | R and S | Urban |
| 1995 | 0.942 | 0.845 | 0.953 | 0.960 | 0.824 | 0.901 | 0.589 | 0.844 | 0.408 | 0.830 | 0.838 | 0.754 |
| 2000 | 0.949 | 0.793 | 0.755 | 0.856 | 0.800 | 0.796 | 0.596 | 0.841 | 0.405 | 0.800 | 0.811 | 0.652 |
| 2005 | 0.942 | 0.853 | 0.807 | 0.960 | 0.768 | 0.879 | 0.605 | 0.852 | 0.402 | 0.836 | 0.824 | 0.696 |
| 2010 | 0.976 | 0.841 | 0.876 | 0.928 | 0.879 | 0.823 | 0.650 | 0.880 | 0.300 | 0.851 | 0.867 | 0.860 |
| Avg. | 0.952 | 0.833 | 0.848 | 0.926 | 0.8178 | 0.859 | 0.61 | 0.8543 | 0.379 | 0.829 | 0.835 | 0.7405 |

R and S: Rural and suburbs.

In Beijing and Tianjin, the "others" type of area has large surface area, and the intensity of the NTL is almost zero, which makes them easier to distinguish. The classification accuracy of the "others" type is the highest with an average of 0.939. Due to the relative concentration of the urban area and the strongest intensity of NTL, the classification accuracy of the urban type is the second best with an average of 0.854. Rural settlements are easy to classify into other areas because their discrete distribution and weak NTL, thus the minimum average accuracy is 0.825. However, with the development of rural areas and the continuous infrastructure improvements, the area and the level of NTL is increasing each year. The retrieval accuracy has also improved, and fewer rural settlements have been left out.

For further validation, we compared the results of quantile methods with the retrieval results of HSI, VANUI, and other indexes based on global-fixed threshold methods, which were acquired from previous studies [27]. As shown in Table 5, the USR retrieval of NTL data using the quantile method achieved relatively good quality results. The OA and KC of Beijing and Tianjin were almost the same as those in the previous method. However, due to the feature of automatic selecting of the optimal thresholds and un-using additional parameters, the retrieval process of the quantile method is more convenient and efficient than are the fixed threshold methods. In addition, we also introduced HSI into the threshold method in the discussion subsection so as to compare the USR retrieval results based on different data (or indicators) through the quantile method.

**Table 5.** Comparison with fixed threshold method.

| | Fixed Threshold Method [27] | | | | | | | | | | Quantile | |
|---|---|---|---|---|---|---|---|---|---|---|---|---|
| Methods | NTL | | HSI | | VANUI | | VTLI | | TVANUI | | NTL | |
| Province | OA | Kappa | OA | Kappa | OA | Kappa | OA | Kappa | OA | Kappa | OA | Kappa |
| Beijing | 0.930 | 0.622 | 0.916 | 0.569 | 0.939 | 0.634 | 0.938 | 0.6283 | 0.941 | 0.681 | 0.912 | 0.669 |
| Tianjin | 0.869 | 0.638 | 0.865 | 0.593 | 0.861 | 0.612 | 0.870 | 0.626 | 0.980 | 0.656 | 0.895 | 0.631 |
| Hebei | 0.909 | 0.625 | 0.904 | 0.567 | 0.908 | 0.621 | 0.906 | 0.613 | 0.909 | 0.640 | 0.625 | 0.383 |

## 5. Discussion

### 5.1. The Problem of Retrieving Rural Settlements

The spatial distribution of rural settlements is more discrete than that of metropolitan areas. The retrieval of rural areas using DMSP/OLS data is limited to indicating the scope of the rural distribution rather than identifying the discrete distribution of rural settlements. Rural settlements will be missed when the following two situations exist:

1. When scattered, a DMSP/OLS sensor cannot detect the NTL intensity, and thus rural settlements will be missed. In addition, due to the blooming effect of NTL, rural settlements cannot be accurately distinguished. Due to the relatively discrete distribution of rural settlements,

when there are many rural settlements, the presence of the blooming effect of the NTL causes the surrounding pixels to be identified as rural areas. As a result, the discretely distributed rural settlements are merged into one area, and so only their approximate scope can be retrieved. Unlike rural settlements, the urban land itself has a more concentrated distribution and larger area. When retrieving the urban area, the impact of the blooming effect is mostly concentrated within the city, and so the impact on the entire urban area is small (Figure 7 Case I).

2.  When using data from earlier years, NTL is not an effective way to identify rural areas, since in the past the economic development in those areas was relatively minor and there was a power shortage in some rural areas. With the construction of the rural infrastructure and the promotion of corresponding policies, the situation of rural electricity consumption has been greatly improved, and the situation of being omitted due to the absence of NTL is gradually reduced (Figure 7 Case II).

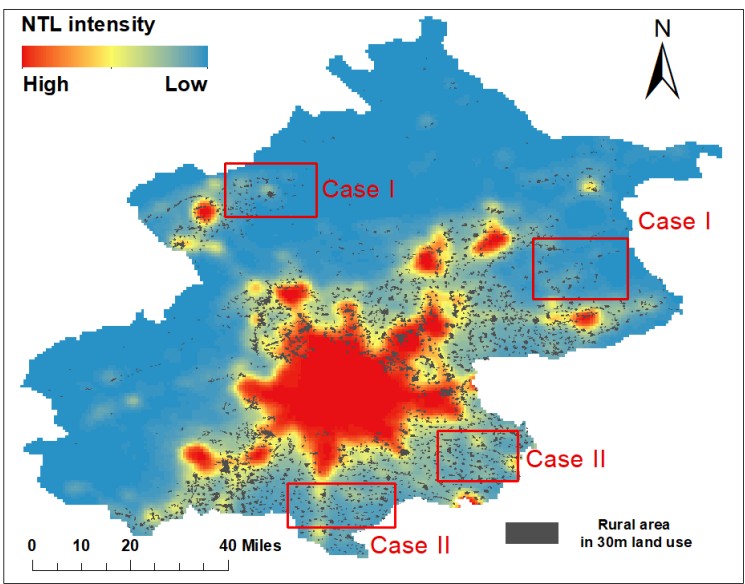

**Figure 7.** Distribution of rural settlements and night lights.

*5.2. Differences between the Results Based on HSI or NTL*

Lu, D et al. proposed the human settlement index (*HSI*) for mapping the extent of the urban area by considering both NTL and vegetation status, which is widely used in mapping the extent of the urban area. The index assumes that the NTL are high and the vegetation index is low in residential areas [1]. In order to compare the impacts of introducing other data or indicators on USR retrieval results, we applied the HSI as an input in the study area of the JJJ. The *HSI*, which combines NTL and *NDVI* data, can be expressed as:

$$HSI = \frac{(1 - NDVI_{max}) + OLS_{nor}}{(1 - OLS_{nor}) + NDVI_{max} + OLS_{nor} * NDVI_{max}} \tag{5}$$

where $OLS_{nor}$ is the normalized night lights value.

After replacing the NTL data with the *HSI* data, the iterative process of the quantile method is performed. Compared with the results retrieved by the quantile method based on the NTL DN values, the OA and KC calculated using the HSI are poor. The results using the HSI are obviously biased in OA, and the average OA in the JJJ is only 0.8. The average KC is 0.485. According to the KC classification standard, the retrieval results only reach general consistency. Different from the NTL-based quantile method, the HSI is less affected by the scale of the study area. In the JJJ, the USR retrieval results of Beijing, Tianjin, and Hebei are more consistent. However, some discretely distributed areas in the city

can be better retrieved. Compared with the results of NTL, the blooming effect is lower (as shown in Table 6). Therefore, it is possible to reduce the problem of excessively large USR retrieval.

**Table 6.** Accuracies for different cities based on HSI.

| Year | 1995 | | 2000 | | 2005 | | 2010 | | Average | |
|---|---|---|---|---|---|---|---|---|---|---|
| Province | OA | KC | OA | KC | OA | KC | OA | KC | OA | Kappa |
| Beijing | 0.892 | 0.543 | 0.901 | 0.560 | 0.779 | 0.491 | 0.821 | 0.455 | 0.848 | 0.512 |
| Tianjin | 0.844 | 0.501 | 0.859 | 0.531 | 0.801 | 0.498 | 0.798 | 0.432 | 0.823 | 0.495 |
| Hebei | 0.724 | 0.435 | 0.764 | 0.467 | 0.755 | 0.451 | 0.712 | 0.433 | 0.739 | 0.447 |

We consider this to have been caused by the shortcomings of the HSI. While using the HSI, the assumption that the relationship between the NDVI and NTL follows the power law is introduced. When NTL reaches its maximum value, the NDVI is close to zero, and the HSI will grow exponentially. Some studies' calculations of the HSI of global cities show that it overcorrects the saturation of urban areas, resulting in a reduction in the data of urban boundary areas [31]. In previous studies, in many regions of the world, urban expansion takes place in areas with reduced or lower vegetation [43]. With the development of China's rural economy and the promotion of the construction of ecological civilization, rural infrastructure has become more complete and urban greening has gradually improved. This makes the difference in the degree of vegetation coverage and the intensity of the NTL between rural and urban areas gradually smaller. The assumption that the HSI retrieves USR is further weakened, which ultimately leads to unsatisfactory results in the final retrieval. This also proves that the retrieval of the structure of USR and the introduction of parameters other than NTL (e.g., VI, LST) will increase the uncertainty in the retrieval results.

*5.3. The Weakness of DMSP and Potential Problems*

DMSP data have some weaknesses, such as bloom effect and oversaturation, which may also reduce the accuracy of the proposed method. Compared with other night light sensors such as VIIRS (750 m spatial resolution) and Luojia (130 m spatial resolution), the spatial resolution is lower [24]. Bloom effect may lead to the combination of lighting in a certain range, which cannot be accurately expressed for the scattered rural areas. Moreover, the problem of oversaturation will gradually blur the boundaries between rural, suburban, and core-urban areas in the process of human activities, which may lead to the disappearance of the difference in NTL intensity of USR in DMSP data. In recent studies, other sensors were processed for spatiotemporal consistency, which adjusted for the defects of DMSP data [44]. For example, by using the data of VIIRS, the problem of merging scattered regions caused by bloom effect can be further suppressed, and there is no oversaturation problem in VIIRS, which can keep the difference of NTL intensity between USR.

In addition, with the development of lighting methods and related technologies, the light radiation also changes in NTL data [45]. Compared to different times, different space ranges at the same time are more important. Table 3 confirms the influence of the space range on the results of the quantile method. For a scale such as the municipal level, the difference in infrastructure construction (such as lights) and human activities is smaller than that of the whole province, and the spatial impact is also relatively small. As for the development of new technologies such as LED, controlling the scope of the research area can better control the lighting difference in the space to reduce the influence of lighting mode on USR retrieval results. However, in the developed countries in Europe or in the United States, there may be counter-urbanization phenomenon, so the retrieval results of USR structure may need to be further modified.

## 6. Conclusions

Considering how few studies previously focused on retrieving the three sub-categories of "core urban-suburban-rural" from nighttime light images, especially suburb areas, we proposed

an improved quantile approach for retrieving the three USR subcategories from NTL images, which automatically defines the boundary thresholds ($D_{rural}$, $D_{suburban}$, and $D_{urban}$) of rural, suburban, and urban areas using DMSP NTL data without introducing empirical knowledge and additional data to retrieve the USR structure. Then, the approach was applied to USR retrieval for the JJJ from 1995 to 2013. According to the retrieval results, the JJJ has experienced rapid urbanization during the past decades. The core urban and suburban areas have increased significantly, and they more than doubled (7098 and 12,690 km$^2$, respectively). Meanwhile, the rural areas expanded at a slower speed by approximately 0.38 times (increase of 4986 km$^2$). In Beijing and Tianjin, the average OA was 0.904 and the average KC was 0.650. Compared with the former method based on a fixed threshold, the quantile method further improves the retrieval accuracy. Through comparison with NTL and the HSI, it was found that NTL DN was more suitable for USR retrieval, being 12.5% on average higher than that of the HSI (from 0.803 to 0.904, respectively). Furthermore, the overall consistency was improved from the general consistency (KC of 0.485) of the HSI to substantial (KC of 0.650) for NTL DN. The retrieval results show that the increase in the parameter uncertainty and the deviation between the original hypothesis and the actual situation may have a serious impact on the retrieval of the USR structure. Due to the blooming effect of NTL, the actual results may overestimate the borders of the countryside, and there will also be missing points for areas that do not meet the assumption of NTL. With the improvement of the spatial resolution of NTL data and sensor sensitivity (such as Luojia No. 1 data), the accuracy of the retrieval of rural settlements with discrete partitions might be further improved. At the city level, the proposed quantile approach achieves similar results to that of the visual interpretation of USR retrieval, with lower labor and time costs, which increases the automation of the method. We believe that the proposed approach will provide an efficient, low-cost method for research on urbanization or urban expansion.

**Author Contributions:** Y.H. and M.C. designed the experiment; C.W. and J.Y. collected and processed the data; Y.H., M.C. and C.W. proposed the model and analyzed the results; Y.H., C.W. and H.R. wrote the manuscript. All authors have read and agreed to the published version of the manuscript.

**Funding:** This work was funded by National Natural Science Foundation of China (Grant NO. 41822104), the National Key Research and Development Program of China (Grant No. 2017YFB0503005 and 2016YFC0401404), the Strategic Priority Research Program of the Chinese Academy of Sciences (XDA19040402) and China high-resolution earth observation system (21-Y20B01-9001-19/22).

**Conflicts of Interest:** The authors declare no conflict of interest.

## Appendix A

**Table A1.** Different land use area for each year.

| Year | Area (km$^2$) | | | | | | | | |
| --- | --- | --- | --- | --- | --- | --- | --- | --- | --- |
| | Beijing | | | Tianjin | | | Hebei | | |
| | Rural | Suburban | Urban | Rural | Suburban | Urban | Rural | Suburban | Urban |
| 1995 | 2092.8 | 2822.9 | 765.8 | 1663.2 | 1789.7 | 385.4 | 9240.6 | 2245.2 | 1384.8 |
| 1995 * | - | - | **1081.8** | - | - | **577.8** | - | - | **1864.5** |
| 1996 | 1805.0 | 2559.7 | 856.6 | 1405.9 | 1597.0 | 386.3 | 10,591.2 | 2385.6 | 1583.4 |
| 1997 | 1580.0 | 2285.5 | 808.2 | 1529.0 | 1321.0 | 438.9 | 10,114.2 | 1906.2 | 1633.2 |
| 1998 | 1816.0 | 2646.3 | 835.4 | 1405.1 | 1467.1 | 444.0 | 10,702.8 | 2645.4 | 1822.2 |
| 1999 | 1608.9 | 2533.4 | 903.3 | 1432.3 | 1600.4 | 476.3 | 8249.4 | 1992.6 | 1852.2 |
| 2000 | 1511.2 | 3301.8 | 1342.3 | 1511.2 | 2333.9 | 483.9 | 9072.6 | 2536.8 | 1892.4 |
| 2000 * | - | - | **1190.3** | - | - | **601.6** | - | - | **1998.0** |
| 2001 | 1462.8 | 3301.8 | 1472.4 | 1511.2 | 2333.9 | 513.3 | 9072.6 | 2536.8 | 1892.4 |
| 2002 | 2272.8 | 2757.6 | 1538.4 | 1233.6 | 2581.8 | 614.7 | 12,408.0 | 2740.2 | 2095.6 |
| 2003 | 2547.0 | 2410.3 | 1601.2 | 4626.2 | 2026.6 | 721.5 | 12,075.0 | 2605.2 | 2101.3 |

**Table A1.** *Cont.*

| Year | Area (km$^2$) | | | | | | | | |
|---|---|---|---|---|---|---|---|---|---|
| | Beijing | | | Tianjin | | | Hebei | | |
| | Rural | Suburban | Urban | Rural | Suburban | Urban | Rural | Suburban | Urban |
| 2004 | 0.0 | 6450.7 | 2013.8 | 0.0 | 3482.3 | 806.6 | 13,365.0 | 2611.2 | 3766.4 |
| 2005 | 0.0 | 5417.5 | 2096.2 | 0.0 | 3960.6 | 656.3 | 11,471.4 | 2298.0 | 3100.1 |
| 2005 * | - | - | **1806.7** | - | - | - | - | - | **2778.3** |
| 2006 | 0.0 | 6980.5 | 2111.5 | 0.0 | 4177.9 | 836.3 | 12,507.6 | 2675.4 | 3138.0 |
| 2007 | 0.0 | 7009.3 | 2279.6 | 0.0 | 4113.4 | 1209.0 | 15,232.8 | 3400.8 | 4460.4 |
| 2008 | 0.0 | 7373.6 | 2419.7 | 0.0 | 4025.1 | 1405.1 | 13,473.6 | 3870.6 | 4661.4 |
| 2009 | 0.0 | 7327.7 | 2483.3 | 0.0 | 3997.9 | 1427.2 | 13,474.2 | 4017.6 | 4323.0 |
| 2010 | 0.0 | 8176.7 | 2816.1 | 0.0 | 4824.0 | 1478.1 | 17,983.2 | 6547.2 | 5340.0 |
| 2010 * | - | - | **2675.2** | - | - | **1167.3** | - | - | **4928.9** |
| 2011 | 0.0 | 6360.7 | 2962.2 | 0.0 | 4521.8 | 1516.3 | 11,727.0 | 4402.8 | 5472.0 |
| 2012 | 0.0 | 7840.5 | 2739.7 | 0.0 | 4670.3 | 1557.9 | 14,831.4 | 4740.6 | 6150.0 |
| 2013 | 0.0 | 7060.3 | 3253.4 | 0.0 | 4892.8 | 1528.2 | 12,919.2 | 5664.0 | 6720.0 |

*: 30 m land use data.

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
