# Peer review of "A Quantile Approach for Retrieving the “Core Urban-Suburban-Rural” (USR) Structure Based on Nighttime Light"

_remotesensing, doi:10.3390/rs12244179_

Round 1
Reviewer 1 Report
The paper proposes a methodology for using nighttime lights to code different types of built-up area: urban, suburban and rural. I think this is something that can be done in principle and I think that the methodology is nice in the sense that it appears to be nonparametric and data driven. There is a substantial literature on using nighttime lights to code built-up area, some of which is cited here. It would be nice for the authors to show more clearly their contribution relative to that literature.
One major comment I had was that the article is very dense and quite difficult to read not just for a nonspecialist but for someone reasonably well familiar with nighttime lights. I would recommend rewriting it for clarity. The authors should explain how their method improves upon the best method in the literature (or if no other paper attempts to distinguish urban from suburban areas, they should state that).
Reviewer 2 Report
The paper deals with a quantile approach for retrieving the core urban-suburban-rural structure based on night-time light. The problem addressed in the paper is interesting, but more details are needed to justify the proposed approach. Also, the results should be better discussed. The details of this revision are reported in the attached file.

Reviewer 3 Report
Figure 4. Minor amendments:
Line 196 : add “(Q)” after “the x-coordinate is the quantile”
Line 197-198 : add “(Q)” after “the y-coordinate is the DN value of the NTL DMSP data”
Line 198 : mark “Figure (d)”,
Lines 224-235
Validation algorithm. What is an "algorithm"? The methods used for verification are not clearly described. Table 2 only confuses a reader.
It is necessary:
- describe the method for calculating the Cohen's Kappa coefficient [Cohen, Jacob. "A coefficient of agreement for nominal scales."Educational and psychological measurement 1 (1960): 37-46],
- define the variables,
- explain the term "overall accuracy (OA)".
Reviewer 4 Report
The paper uses DMSP for urban studies, a widely used method.
The paper provides a new thresholding using a quatile approach rather then smple thresholding.
It is outperforming the standard method for one urban region but produces about the same result for 2 others.
Generally the method provides a small step and authors have justify better why this study is important. Many points seem to be in the manuscript but they should be ordered better and brought more to the point.
It is not the most modern sensor (DMSP) and a discussion on how this method could be applied to more modern systems (VIIRS and Luojia etc.) is missing.
However, the paper in its present state is not ready for publication yet.
Abstract is not clear
Motivation is not clear.
The weaknesses of DMSP are not clearly discussed.
Here are further comments:
Abstract
An abstract must be self-explanatory and include all necessary information
It needs to be improved.
There are too many abbreviations in the abstract
BMA is not properly introduced (I guess it is Beijing metropolitan area)
I find the abbreviation CSR confusing - for me this is linked to "corporate social responsibility (CSR)"
the abbreviation for urban suburban interfaces seems uncommon to me
Introduction
line 38: there are certainly more implications of urbanization than the two listed - to list all of them is exhaustive but maybe emphasize this fact that it is one of the grand challenges of humanity
line 40: is there also an English version of ref. [2]?
or maybe this one fits as well (?): Farrell, K. The Rapid Urban Growth Triad: A New Conceptual Framework for Examining the Urban Transition in Developing Countries. Sustainability 2017, 9, 1407.
line 50 onwards: this sentence is hard to read
line 56: features of optical satellite images - maybe add "daytime" (?)
line 65: I think [20] is not about NTL
but maybe add: Levin, Noam, et al. "Remote sensing of night lights: A review and an outlook for the future." Remote Sensing of Environment 237 (2020): 111443.
this gives also a lot of detail on different NTL satellites and other applications of NTL
line 81: good that blooming and saturation is mentioned, but it should be more emphysized that this is a weakness of DMSP
line 96: please try to formulate the deficiencies of past research and the need for your research clearer
line 100: now BMA is introduced but not in abstract
line 103: the last senetnce can be omitted
Dataset and Study area
I think this can be combined in Methods but its not a big deal.
line 125 and following: I think this needs to go into the introductions as the land use conflicts and interface problem is a motivation for the study
line 137: DMSP NTL data varies a lot over time but there are strategies to overcome this - it is not clear which correction for the temporal trends have been used
authors also should clarify why this time frame was used
Methods
I think the methods section could be expanded a little bit discussing the downsides of DMSP, this info is scattered around in the manuscript and could be combined somewhere
line 178: is NTL intensity linear or log scale in the figures?
Results
line 225: please specify "visual interpretation"
line 255: maybe give a table for the different land use classes for ecach year - could be in appendix
line 286: table 5 - maybe rather plot the differences or produce a heatmap to show where the new method performs better?
line 287: if the threshold method perfomrs almost equally in Beijing and Tianjin please discuss reasons for this better and also discuss why the novel method did perform so much better in Hebei
Discussion
Please discuss changes in lighting technology. Was the same lamp type used? What lamp type is normally used in theses areas? Is it high proessure sodium? Or diufferent light?
Also LED light could mask results because different lamp type produces different upward radiance - and therefore different NTL signal
see Fig. 5 and discussion around this in
Kyba et al. "Artificially lit surface of Earth at night increasing in radiance and extent." Science advances 3.11 (2017): e1701528.
In US and European cities this should only matter after the period investigated here.
But maybe there were unexpected changes in the core region that could indicate early use of LED?
line 300: what does "when independent" mean? I do not understand
you mean the sensitivity is too low?
Figure 7: how is rural area defined or acquired? - check spelling in figure
line 316: maybe a figure subtracting the land use classes obtained from eq. 1 (HSI) and the newly proposed NTL method is helpful?
Conclusion
needs to be rechecked after revising the rest
I am also missing how this can be used in the future - VIIRS could give calibrated radiance but Luojia is not continuosly operating
So please discuss
line 366: resolution of NTL - please specify "spatial resolution"
References
several references are incomplete e.g. missing page information, please check
Round 2
Reviewer 1 Report
The paper has addressed the comments that I had and is fine for publication.
Reviewer 2 Report
The paper has been reviewed according to the provided suggestions. The content of the manuscript is interesting and worthy of publication.
Reviewer 4 Report
authors have addressed all my points
I only find the change from BMA to JJJ even more confusing ;)
maybe authors rethink this change